# A Critical Review on the Influence of Fine Recycled Aggregates on Technical Performance, Environmental Impact and Cost of Concrete

**Hisham Hafez [1],*, Reben Kurda [2], Rawaz Kurda [3],* , Botan Al-Hadad [4], Rasheed Mustafa [5] and Barham Ali [6]**

[1] Department of Mechanical and Construction Engineering, University of Northumbria, Newcastle upon Tyne NE18ST, UK

[2] Department of Information System Engineering, Erbil Technical Engineering College, Erbil Polytechnic University, Erbil, Kurdistan Region 44001, Iraq; Reben.kurda@epu.edu.iq

[3] Department of Civil Engineering, Technical Engineering College, Erbil Polytechnic University, Erbil, Kurdistan Region 44001, Iraq

[4] Erbil Technology Institute, Erbil Polytechnic University, Erbil, Kurdistan Region 44001, Iraq; botan@epu.edu.iq

[5] Department of Environmental Engineering, College of Engineering, Knowledge University, Erbil, Kurdistan 44001, Iraq; rasheed1954@yahoo.com

[6] Civil Engineering Department, Engineering Faculty, Tishk International University, Erbil, Kurdistan Region 44001, Iraq; barham.haydar@tiu.edu.iq

\* Correspondence: hisham.hafez@northumbria.ac.uk (H.H.); rawaz.kurda@epu.edu.iq (R.K.); Tel.: +44-7493280699 (H.H.); +351-965722755 (R.K.)

**Abstract:** The aim of this critical review is to show the applicability of recycled fine aggregates (RFA) in concrete regarding technical performance, environmental impact, energy consumption and cost. It is not possible to judge the performance of concrete by considering one dimension. Thus, this study focussed on the fresh and hardened (e.g., mechanical and durability) properties and environmental and economic life cycle assessment of concrete. Most literature investigated showed that any addition of recycled fine aggregates from construction and demolition waste as a replacement for natural fine aggregates proves detrimental to the functional properties (quality) of the resulting concrete. However, the incorporation of recycled fine aggregates in concrete was proven to enhance the environmental and economic performance. In this study, an extensive literature review based multi criteria decision making analysis framework was made to evaluate the effect of RFA on functional, environmental, and economic parameters of concrete. The results show that sustainability of RFA based concrete is very sensitive to transportation distances. Several scenarios for the transportation distances of natural and recycled fine aggregates and their results show that only if the transportation distance of the natural aggregates is more than double that of RFA, e the RFA based concrete alternatives would be considered as more sustainable.

**Keywords:** fine recycled aggregates; construction and demolition waste; recycled aggregate concrete; life cycle assessment; sustainability; optimization

## 1. Introduction

Due to the rising need for urbanization, concrete usage is expected to be doubled by 2050 [1]. Given the fact that aggregates comprise around 70% by volume of concrete, alarming depletion rates of non-renewable natural resources such as river sand and limestone are expected [2]. Already, there

are more than 30 billion tons of natural aggregates consumed annually for producing concrete globally. The average increase in aggregates use is estimated at 5% per year globally, which means that the total could be around 66.3 billion tonnes by 2022 [3]. According to study [4], recycled aggregate (RA) still represents only three percent of the total aggregates demand. In Europe, construction and demolition waste (CDW) occupies one third of the waste produced, but significant differences can be found in the level of recycling among different countries [2]. Similarly, the construction industry is the second largest in generating waste, either during construction or demolition [5]. Four billion tonnes of CDW are generated in China yearly [6] and around 900 million tonnes in Europe [7]. Surprisingly, only 5% of the currently generated CDW is being recycled in concrete in any form [4]. Accordingly, special attention is devoted to the potential of recycling CDW as aggregates in concrete to simultaneously reduce the threat of depleting natural resources and avoid the landfilling of CDW [2,8,9].

In line with the European Union Directive [10] that encourages the reuse and recycling of waste materials, researchers examined the feasibility of creating sustainable concrete by replacing naturally sourced aggregates (NA) in concrete with recycled aggregates from CDW in terms of mechanical properties, durability, environmental impact (EI) and cost [11–16]. Generally, it was agreed among researchers that in addition to the potential of being a cheaper alternative, producing recycled aggregates concrete (RAC) could reduce the EI of concrete specifically in terms of "landfill use". However, it was also concluded that the technical performance would be adversely affected. Using RA, instead of NA, leads to a reduction of the mechanical properties and durability of a RAC compared to conventional concrete [17–21].

Regardless of the numerous studies found, there was no consensus found on the replacement ratio of fine NA by fine RA (RFA) from CDW that would achieve the optimum sustainability-potential of the resulting concrete mix. The sustainable development potential of a material depends on three main parameters: its functionality, its EI and economic viability [22]. Therefore, this paper will attempt to assess the optimum sustainability potential, based on a multi criteria decision analysis framework, which combines these three sources. However, the scope of the paper will only focus on RFA from CDW (RFA) being incorporated, with different percentages, in concrete replacing naturally sourced fine aggregates. Recently, there was a shift in the direction of the research concerning recycled aggregates focusing more on RFA made with glass, lightweight materials, etc. The reason is that these studies [23–39] recommended not to include CDW as RFA anymore just on the basis of its functional properties. For that reason, in this study, the applicability of RFA in concrete in terms of all parameters (quality, environmental impact, energy consumption, toxicity and cost) was considered. Regression models were developed based on data obtained from the literature, and then, a multi criteria decision making analysis framework was used to combine the results for the mentioned parameters and to find the optimum incorporation level of RFA in concrete. However, it is important to highlight that the statistical assumption that the raw data used in the regression models follow a normal distribution was not verified. The reason is that the sample sizes were too small to eliminate any of the data as will be shown later.

## 2. State-of-Art Review

### 2.1. General Parameters Affecting the Performance of RFA in Concrete

#### 2.1.1. Recycling Process

The process by which CDW is turned into aggregates, whether coarse or fine, suitable for reintegration in concrete can be divided in several stages, namely separation by metal detectors, initial screening, crushing, secondary screening, secondary crushing and then sieving and separating [40]. This number of stages of crushing has a significant effect on the quality of the RFA produced and subsequently on the RAC which will include it. Source concretes subjected to a secondary crushing procedure in an impact crusher normally result in RFA with less attached mortar than those following only a primary crushing procedure. Thus, due to the hardened cement paste having higher porosity

than that of NA, as the content of adhered-mortar increases, so does the RA's water absorption [41]. Kim and Yun [42] obtained RFA from demolished concrete of current compressive strength (21 MPa). By repeatedly crushing and sorting, the authors used water absorption ratios of 6% and 8% to evaluate the effect of water absorption ratio of the RFA on the bond strength between reinforcing bar and RAC. Lee [43] investigated two different crushing processes using a jaw crusher and an impact crusher. The work resulted in oven-dried density of 2390 kg/m$^3$ and 2280 kg/m$^3$ and water absorption of 6.59% and 10.35%, respectively. The process of washing the RA also has a strong effect on their WA. A study by Wegen and Haverkort [44] showed that, after RA were washed, the water absorption of the aggregates dropped by 35% to 55%. This can be explained by the fact that very small particles were removed by the water after washing the aggregates, which significantly affected the water absorption of the fine RA. Song and Ryou [45] washed fine RFA with different washing stages and used a combination of chemical and physical processes. The process of washing the RFA had several effects; the water absorption fell from 6% to 2%, the impurity content fell from 0.5% to 0.2% and the ratio of absolute volume increased from 62% to 65%. Hence, it could be concluded that it is advisable to use jaw crushers rather than impact ones and to do several cycled of washing and crushing the CDW. However, it should be noted that the more cycles done, the higher the energy use is, which will then reflect on the environmental performance of the resulting RFA as discussed in Section 2.3.

### 2.1.2. Particle Size

The recycling process though as explained in Section 2.1.1 could be very thorough, it does not detach the old mortar from the aggregates. Hence, as seen in Figure 1, NA have only one transition zone between the aggregate and the cement matrix, and it is a porous narrow band which forms at the cement past/aggregate interface called the interfacial transition zone (ITZ). For RFA there are two ITZ phases: a new ITZ between the new cement paste and the old mortar from the RFA and an old ITZ between the old cement paste and the original aggregate NA [46]. Moreover, the crushing process itself results in micro cracks in the adhered mortar/original aggregate. These pores and cracks in the RFA absorb water and lead to high water content, which then leads to weaker concrete.

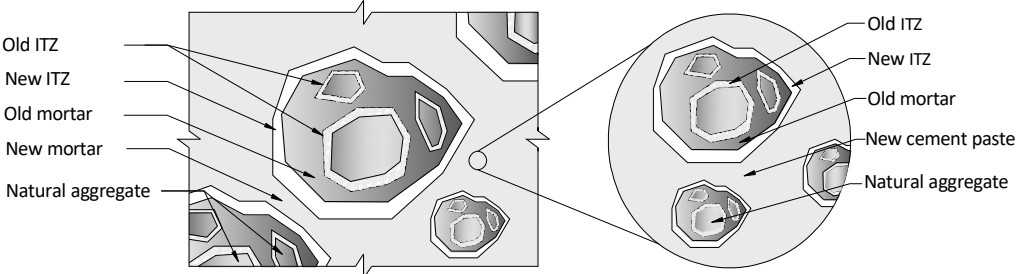

**Figure 1.** Structure of full replacement of fine and coarse naturally sourced aggregates (NA) with fine and coarse recycled aggregate (RA) and their transition zones.

According to study of Lima et al. [47], the higher the minimum particle size of the RFA is, the higher it would contain old adhered cement binder. Fineness modulus is a standard index that defines how fine or coarse a sample of sand is. A fineness modulus value smaller than 2.8 indicates fine sand, while a larger number indicates coarse sand. Evidence shows that the fineness modulus of the RFA incorporated in concrete has a significant effect on several functional parameters. First, for RFA with similar oven-dried density and lower fineness modulus (<2.8), the following studies [15,47–52] obtained higher water absorption compared to the results of the studies, which used RFA with higher fineness modulus (>2.8) [42,53–66]. Similarly, Geng and Sun [67] studied the carbonation behaviour of RFA concrete with different minimum particle sizes and incorporation level of RFA with two design compositions. In the first design, the water content was kept constant and in the second design the workability was kept constant. The authors reported that as the minimum particle sizes of RFA

decreases, the carbonation depth of RFA based concrete increase. Contrary to the previous statement, Geng and Sun [67] studied the effect of RFA's particle size and reported that, as seen in Figure 2, the compressive strength of concrete for the same incorporation levels of RFA decreased with increasing content of smaller fine RFA particles. The researchers cited the fact that larger RFA particles have higher adhesion properties due to the presence of a higher % of old adhered binder.

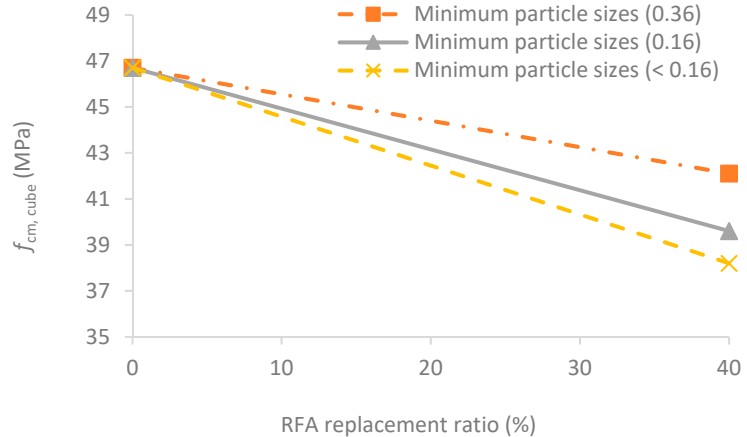

**Figure 2.** Influence of minimum fine recycled fine aggregates (RFA) particles size on 28 days strength of concrete [67].

### 2.1.3. Quality of the Source Material

Solyman [66] analysed the influence incorporation of different types and levels of fine RA on concrete's modulus of elasticity and workability of the resulting RAC. As seen in Figure 3, among 9 different types of CDW, it was concluded that using 80% old concrete +20% old bricks as RFA yields the least drop in workability and in elastic modulus with the higher incorporation rates. The reason could also be related to optimizing the water absorption potential.

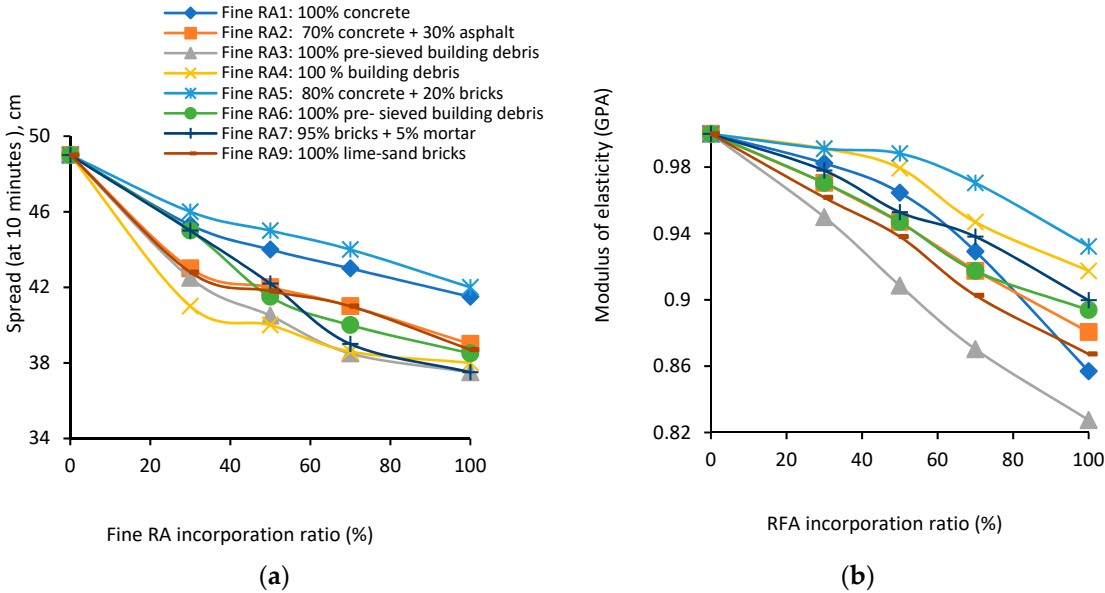

**Figure 3.** Effect of replacing different incorporation ratios and types of fine RA content on (**a**) workability and (**b**) relative modulus of elasticity of concrete [66].

## 2.2. The Effect of Varying RFA Incorporation Ratio on Concrete Performance

### 2.2.1. Workability

Figure 4 presents the relationship between the concrete slump and the incorporation levels of RFA, sourced from these studies [64,65,67]. A reasonable explanation for this may be that RFA absorbs more water from the mix than NA [47,68,69]. Moreover, the minimum particle size of RFA affects concrete's workability because smaller RFA have higher water absorption than the bigger ones from the same source concrete. Greater surface-area and higher water absorption of the fine particle sizes of RA have more potential for higher water-demand and decrease workability as a result [67]. The RFA fineness level and its effect on workability are other aspects that should be considered.

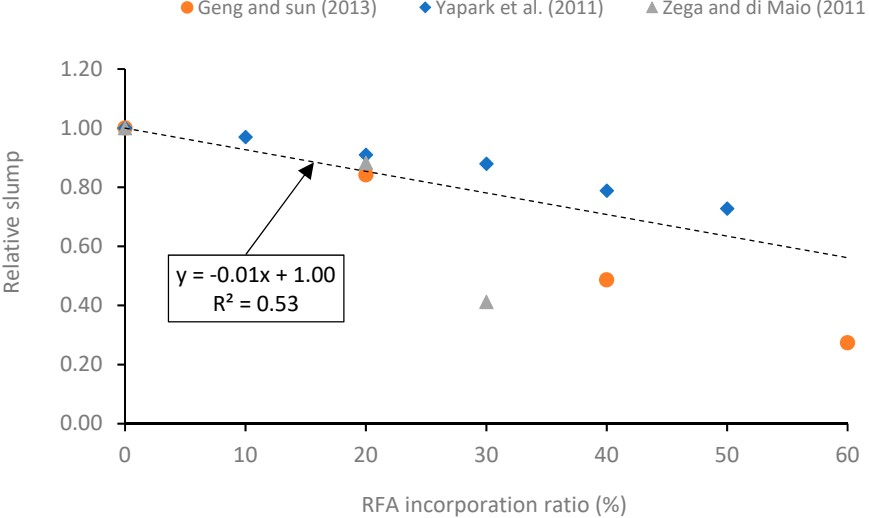

**Figure 4.** Relationship between RFA concrete and slump for similar w/c ratio.

Yaprak et al. [64] obtained slump values between 85 and 165 mm for fresh concrete that were made with different incorporation levels of RFA. The data of this study confirm that the slump of RFA concrete decreases as the incorporation level of RFA increases. This discrepancy might be due to the texture, shape, water absorption and dust content of the crushed RFA when compared to the natural sand. Geng and Sun [67] kept the water/binder (w/b) ratio constant at 0.40 and reported that RFA based concrete has a higher water adsorption leading to poor slump and workability at higher RFA incorporation ratios.

For normal RFA concrete without using superplasticizers (SP), these studies [54,70,71] showed that the w/c ratio needs to be increased as the incorporation ratio of RFA increased in the concrete mixes to get the same target slump. This could be achieved by adding 15% extra water to the RFA concrete [72]. The following studies by [73–76] confirmed this trend. Nevertheless, Leite [50] reported that the workability of concrete with high fineness of RFA may not decrease. According to Figure 4, it could be concluded from the literature that given a concrete mix has a starting value for slump equal to $S_i$, the incorporation of a X% of RFA as a replacement to fine NA would result in a slump value of $Y_{slump}$ in Equation (1).

$$Y_{slump} = Si \times (1 - 0.05X) \tag{1}$$

### 2.2.2. Bulk Fresh Density

Generally, the density of RFA concrete is lower than that of the conventional concrete and their density decrease with increasing incorporation of RFA due to the presence of lower density residual-cement mortars attached to aggregate particles [73]. Figure 5 shows the influence of increasing incorporation of RFA on the bulk density of concrete mixes in the fresh state, sourced from these

studies [54,64,70,71,77]. In order to avoid sacrificial pseudo-replication error [78], for each study, the data of high- and low-strength concrete mixes are separated then the global trends for each concrete are drawn. The fresh-bulk density of low-strength concrete mixes were lower than that of high-strength concrete mixes, and the fresh density of both concrete types decreased when replacing fine NA with RFA. The results show that the rate of reduction in the fresh density by increasing incorporation of RFA was similar for high- and normal-strength concrete.

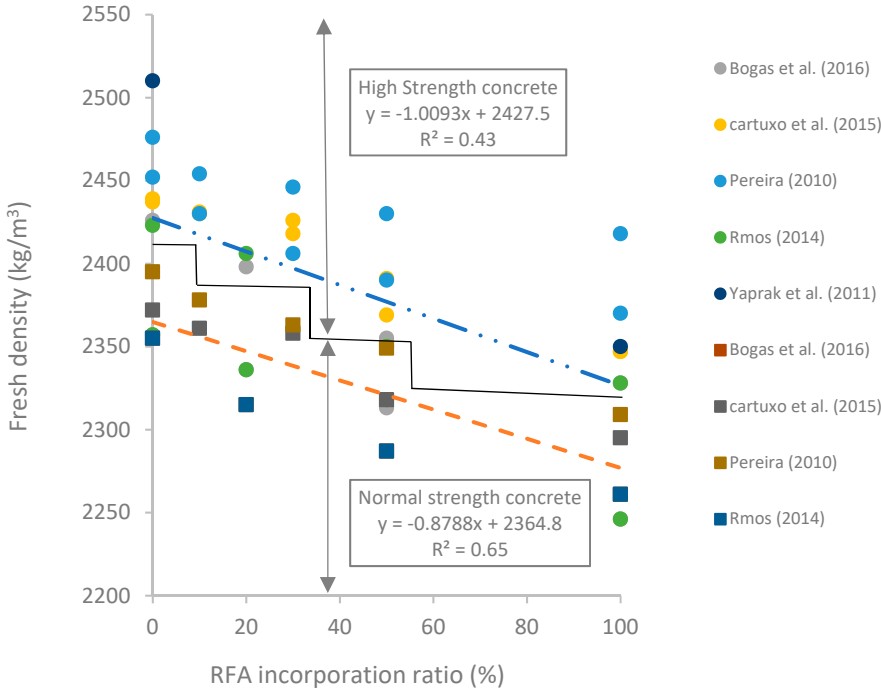

**Figure 5.** The effect of incorporating RFA on the fresh bulk density of high- and normal-strength concrete.

Bogas et al. [77] and Ramos [71] obtained similar results and both investigated three concrete mix families. The first family was normal-strength concrete without admixtures, the second family was high-strength concrete without air entrainment, and the third family was high-strength concrete with air entrainment. The fresh density of the three families decreased between 1% and 5% as the replacement ratio of fine NA with RA increased. The reduction range due to incorporating RFA did not change for high- and low-strength concrete. Moreover, the use of SP increased the fresh density of concrete by about 3% as a consequence of lowering the w/c ratio and increased the compactness of concrete as a result, while the use of air enter admixture offset the gain in density obtained by using the SP. Yaprak et al. [64] obtained unit weight values of fresh concrete mixes. The values changed between 2350 kg/m$^3$ and 2510 kg/m$^3$. The unit-weight value decreased as the RFA content in the concrete increased. The reason behind this was that the specific gravity of RFA was lower than that of fine NA.

### 2.2.3. Compressive Strength

The compressive strength of concrete is one of the most significant parameters that determine the capacity of a structural member to withstand certain loads in service. The compressive strength of RAC normally depends on the age of concrete, RFA incorporation level, additives, admixtures, w/c, quality of the source material and moisture content, type, and size of the RFA [79]. There are two contrary results drawn from different researches work on this subject. Figure 6 summarizes the results of the following studies and shows the influence of increasing incorporation of RFA on the compressive strength of concrete mixes over time. The majority of the researchers arrived to the same conclusion, i.e., the incorporation of RFA in concrete mixes is harmful in terms of compressive strength [49,64,67,71,77,80]. This is related to several factors, the most effective being the water content

needed to increase the w/c in RAC mixes to archive the same workability to that of NA concrete. This can be related to the high-water absorption of the RFA and its texture and angular shape. Nevertheless, the results of these studies [15,42,70,71] show that RFA does not significantly influence the compressive strength of the low-strength concrete mixes [15,42,70,71] because the ultimate strength of low-strength concrete mainly depends on the quality of paste rather than aggregates.

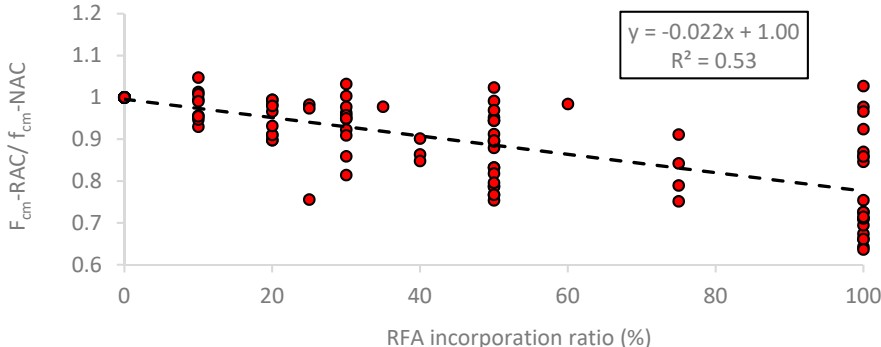

**Figure 6.** Effect of incorporation of RFA in concrete on compressive strength over time.

Evangelista and de Brito [15] reported that the compressive strength of concrete with up to 30% of RFA may not be jeopardized. In addition, the loss for 100% of RFA was only 7%. This behaviour can be explained by the fact that RFA increased the total cement content (during the crushing process of source concrete, some hydration products that trapped the original cement particles may break and release the non-hydrated cement particles), which can reach as much as 25% of its weight [81]. Kou and Poon [49] have demonstrated that replacing 25% to 50% of the fine NA with RFA does not affect the compressive strength of self-compacted concrete. Ahmed [82] confirmed that the compressive strength of concrete with replacement of up to 50% of fine NA with RFA was similar to that of the reference concrete, or even slightly higher. For higher replacement ratios, the maximum loss of compressive strength was 13% and 22% at 28 and 56 days, respectively. The results of Leite's [50] study show that the compressive strength of concrete mixes increased as the incorporation levels of RFA increased. The researcher believes that concrete gained strength by incorporating RFA because the roughness of RFA is bigger than that of fine NA. This causes a better bond and increases the stiffness of cement paste because RFA has more porosity than fine NA, and it allows the acceleration of cement hydration crystals in its pores. To conclude, although with a low statistical significance, based on the aforementioned literature findings, it could be generalized that if a mix has an initial compressive strength value equivalent to $F_{ci}$, the replacement of fine NA with a percentage X of RFA results in a strength Y in Equation (2).

$$Y = F_{ci} - 0.022X \qquad (2)$$

2.2.4. Modulus of Elasticity

Several investigations found a relationship between the modulus of elasticity of RFA concrete and its compressive strength. Based on these results, the authors of this study have derived another equation (Figure 7). There is a gradual increase of the modulus of elasticity of RFA concrete when its compressive strength increases for normal-strength concrete, but for high-strength concrete, the graph shows that there is only a slight increase when compressive strength increases. Hence, the regression model in Table 1 was established.

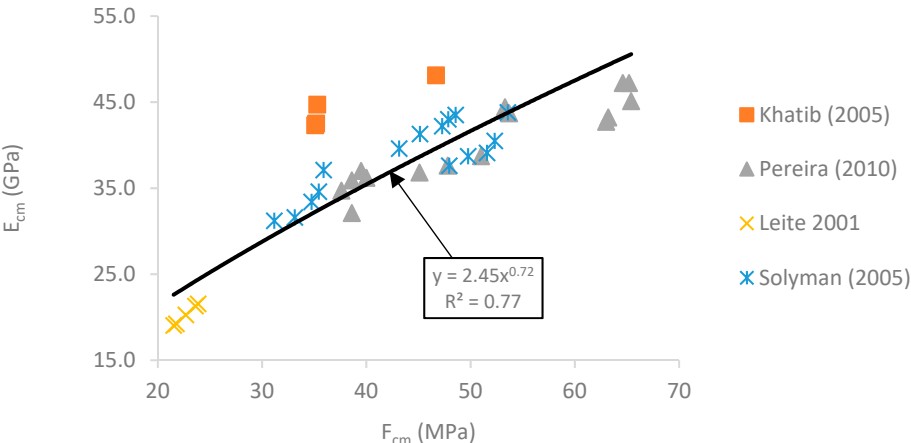

**Figure 7.** Relationship between compressive strength and modulus of elasticity for increasing incorporation levels of RFA concrete.

**Table 1.** Relationship between modulus of elasticity and compressive strength of RFA concrete according to various equations.

| Equation | Notes | Reference |
|---|---|---|
| $E_{cm} = a \cdot f_c^{\frac{1}{3}} \cdot ((1-r) \cdot \rho_{FNA} + r \cdot \rho_{FRA}) \cdot \left[ \frac{\left( \frac{w}{c} \right)_{RC0}}{\left( \frac{w}{c} \right)} \right]^b$ | Taking into account the w/c ratio; replacement ratio; quality RFA. Effective w/c ratio considered 0.55 and correlation factors a and b equal to 4.228 and 0.22, respectively, with a coefficient of determination $R^2 = 0.916$. The equation was developed based on Model code [83] | [84] |
| $E_c = 2.58 \, f_{ck}^{0.63}$ | Relationship between compressive strength and modulus of elasticity. The equation can be used to determine RFA and CRA concrete, taking into account the modulus of elasticity of RFA concrete 10% lower than that of CRA concrete | [85] |
| $Y_E = 2.58 - 0.0159 F_c^2 + 1.9107 F_c - 13.276$ | The relationship between compressive strength and modules of elasticity. With a coefficient of determination $R^2 = 0.80$ | Literature review |

### 2.2.5. Carbonation Resistance

The content and type of aggregates affect the pores system of hardened concrete. Generally, it is agreed that RA's incorporation decreases the density of concrete because of the adhered mortar. Thus, it can be said that RAC has more pores than NAC, which allow the atmospheric $CO_2$ to diffuse into the hardened concrete more easily and, when the carbonation reaction takes place, the alkalinity of concrete reduces (Figure 8). Moreover, the influence of RAC on carbonation of concrete may depend on the incorporation level, crushing-procedure and quality of the RA, and concrete exposure to different curing conditions, admixture use, degree of hydration over time, and mineral additions use [79].

There are grounds to suppose that carbonation increases as the replacement level of fine NA with RFA increases when the binder content is constant in the concrete mixes. The following studies [67,86] all agreed with the previous supposition. The carbonation depth increases with the porosity's increment [87]. As the incorporation level of RFA increases, the porosity of RFA concrete increases because the dry particle density of RFA is much lower than that of fine NA due to adhered mortar. Broadly speaking, it can be said that the pores of RFA are much greater than those of fine NA. However, Levy and Helene [88] obtained lower carbonation depth for 20% and 50% replacement of fine NA with RFA. The authors reported that this can be explained by increased cement content to achieve the same compressive strength of NAC. But for 100% replacement of fine NA with RFA, carbonation increased even when the cement content increased in the mixes (Figure 9).

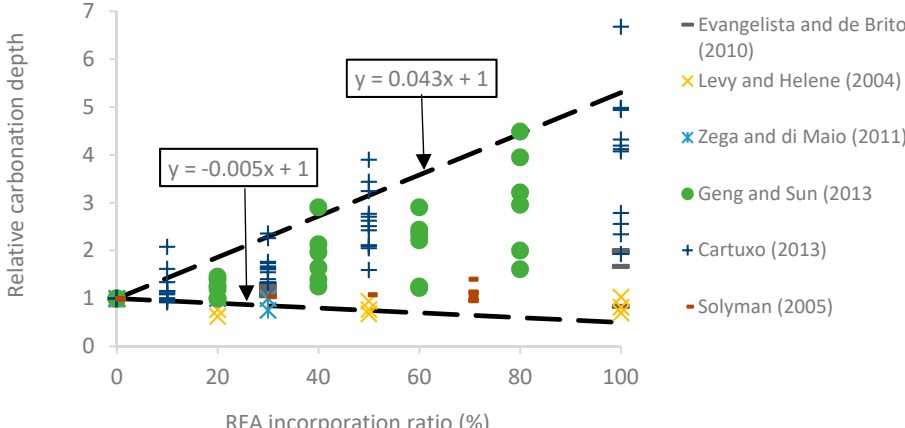

**Figure 8.** Detail of different particles densities in the same RAC mix.

**Figure 9.** Relative carbonation depth for various RFA incorporation levels regardless of the age of mixes.

Figure 10 shows that carbonation increased as the replacement ratio of fine NA with RFA increased while for the same mixes compressive strength decreased. Equation (3) can be deduced based on the relationship between RFA incorporation and concrete compressive strength and carbonation with a coefficient of determination $R^2$ equal to 0.83.

$$Carbonation\ depth_{fine\ RCA} =\ Carbonation\ depth_{Ref.} \cdot \left(18.273 \cdot \left(\frac{Fcm_{fine\ RCA}}{Fcm_{Ref.}}\right)^2 - 43.301 \cdot \frac{Fcm_{fine\ RCA}}{Fcm_{Ref.}} + 26.092 \right) \quad (3)$$

The data seem to suggest that carbonation depth for fixed w/c ratio steadily increases up to 40% of replacement of fine NA with RFA. For higher values, the trend shows steep increases in the carbonation depth. For constant workability, the carbonation depth of concrete mixes increased at least 16.5% with increasing incorporation of RFA at 28 days. The water content is one of the main factors that affects RFA concrete's carbonation. Therefore, the w/c ratio should be considered at first, especially when the incorporation level of RFA exceeds 40%.

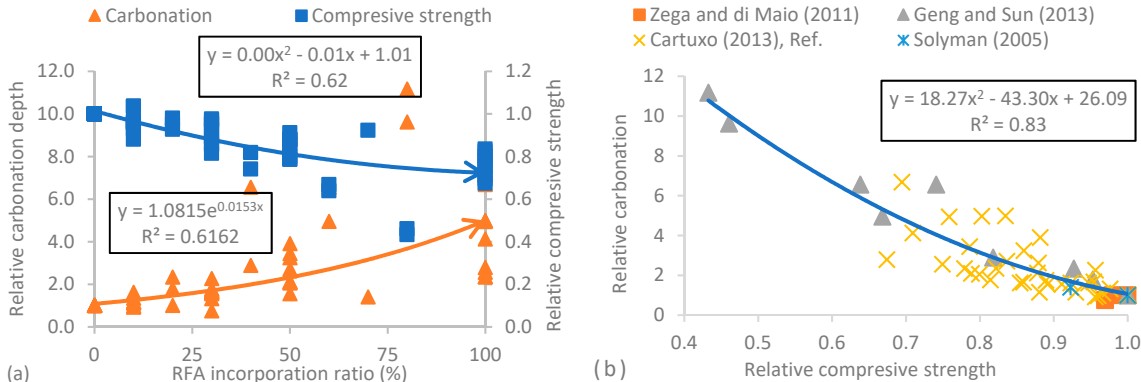

**Figure 10.** Relative carbonation of RA concrete vs. (**a**) RFA incorporation level and concrete relative compressive strength (**b**) relative concrete compressive strength.

Evangelista and de Brito [89] measured the carbonation depth at different ages for concrete samples made with 30% and 100% of RFA. Their study confirms that carbonation depth increases as the incorporation of RFA increases. The carbonation depth at 14, 21 and 91 days increased by 100%, 100% and 70%, respectively, when fine NA were fully replaced. From the above observations, it follows that the carbonation rate of concrete with RFA relative to conventional concrete decreases for later ages (Figure 11). Hence, given a conventional concrete mix has an expected 90 days carbonation depth of $D_{90D}$, the mixes with incorporation % X would have a carbonation depth $Y_{cd}$ in Equation (4).

$$Y_{CD} = D_{90D} + 0.041X \qquad (4)$$

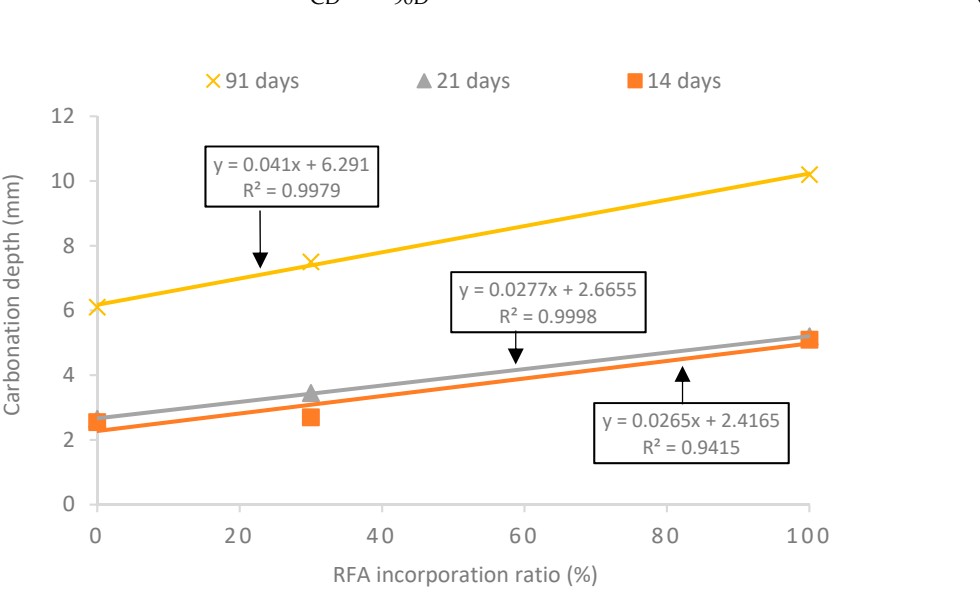

**Figure 11.** Carbonation depth over time of concrete mixes versus fine NA with RFA incorporation ratio.

### 2.2.6. Chloride Penetration Resistance

The chloride ion penetration resistance of concrete is highly affected by the pore content in concrete, similarly to carbonation resistance. Generally, it can be said that as the incorporation of RFA increases the chloride penetration resistance decreases because RFA increases concrete's porosity. The source of the deleterious action of chloride ions is generally classified into two types: internal agents, such as the mixing water, aggregate or admixtures, e.g., in very low temperature calcium chloride is used as an accelerator, and agents that may come from external sources, e.g., marine environment, pools and de-icing salts [90].

In his study, Silva [79] listed several factors that affect the chloride ion penetration of RAC, such as the RA replacement level, quality of the RA, exposure to different curing conditions, mixing procedure, admixture use, and use of mineral additions. The author reported that RA leads to higher chloride ion penetration depths than NA. The aggregates subjected to primary plus secondary crushing procedure are better than those from primary only crushing procedure because aggregates subjected to secondary cursing procedure have lower adhered mortar content, which leads to lower aggregate porosity. Thus, the RA allow lower permeability and chloride ion penetration. The resistance of RAC to chloride ion penetration approaches the resistance of NAC with the passage of time. The addition of materials and steam curing have a positive effect on RAC performance in terms of chloride ion penetration. The following studies [86,89] concluded that incorporating 10% of RFA may have a positive effect on the resistance to chloride ion penetration. For higher values, similarly to carbonation, the chloride ion penetration resistance decreases as the incorporation of RFA increases (Figure 12). The regression model displayed in the graph shows that given a conventional concrete mix exhibiting a certain chloride penetration potential $P_{Cl1}$, the predicted value is relative to that with the addition of a % X of RFA $P_{Cl2}$ (Equation (5)).

$$Y_{cl} \ (P_{Cl2}/P_{Cl1}) = 1 + 0.0033X \tag{5}$$

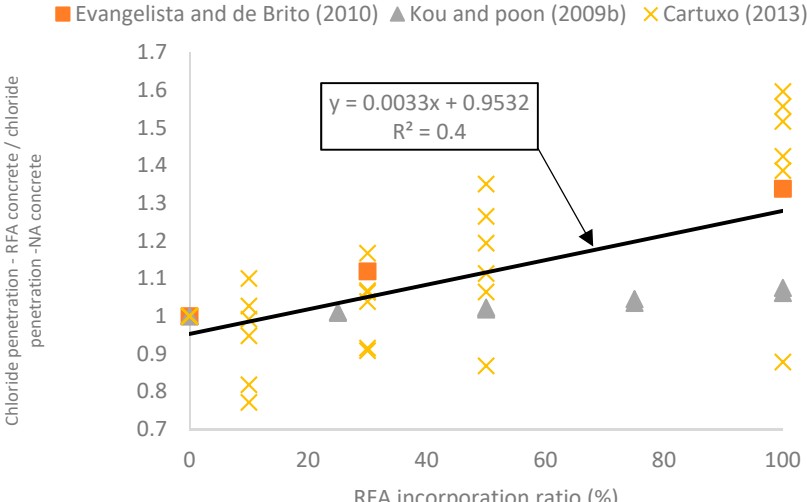

**Figure 12.** Influence of increasing the RFA content on chloride penetration resistance.

## 2.3. The Effect of Varying RFA Incorporation Ratio on Environmental Impact of Concrete

### 2.3.1. Toxicity of Raw Materials for Concrete

An established method to assess the risk of toxicity of construction materials is the quantification, using standard tests, of the possibility of leaching potentially harmful substances to the environment. An indication would be to test the traces of heavy metals, TOC (total organic carbon), the phenol index (carboxyl, halogen, hydroxyl, methoxyl or sulfonic acid), NOx, SOx, and the pH. Regarding the recycled aggregates from CDW, Barbudo et al. [91] concluded that aggregates from CDW are only susceptible to minimal heavy metal leaching. However, high concentrations of $SO_3$ compounds, which can cause the pollution of superficial and/or ground water, were found in mixed RA containing either ceramic particles or gypsum. Another study [92] on the leaching characteristic of unbound RFA showed that "leached heavy metals did not exceed the Norwegian drinking water criteria".

### 2.3.2. Life Cycle Assessment of RFA in Concrete

A life cycle assessment (LCA) is an agreed methodology to identify and quantify the EI and energy consumption paving the way for making improvements to the environmental sustainability of the process or product understudy [93,94]. An LCA can be classified into 4 stages: (1) the system

boundary. A system boundary of a concrete product could be Cradle-to-Gate, which means including all processes and emissions until the production of its different constituents or Cradle-to-Grave which includes the "Use" and "End-of-Life" phases as per Table 2 [95]. In most of the references reviewed in this paper, a cradle-to-gate boundary was selected on the strict condition that the CDW are allocated the avoidance of landfill from their previous life cycle.

**Table 2.** Life cycle assessment (LCA) system boundary.

| LCA Boundaries | Life Cycle Stages | | Life Cycle Stage |
|---|---|---|---|
| Cradle to gate | Product stage (A1–A3) | A1 | Raw material extraction and processing, processing of secondary material input |
| | | A2 | Transport to the manufacturer |
| | | A3 | Manufacturing |
| Gate to grave | Construction process stage (A4–A5) | A4 | Transport to the building site |
| | | A5 | Installation into the building |
| | Use stage—information modules related to the building fabric (B1–B5) | B1 | Use or application of the installed product |
| | | B2 | Maintenance |
| | | B3 | Repair |
| | Use stage—information modules related to the operation of the building (B4–B5) | B4 | Operational energy use |
| | | B5 | Operational water use |
| | End-of-life stage (C1–C4) | C1 | De-construction, demolition |
| | | C2 | Transport to waste processing |
| | | C3 | Waste processing for reuse, recovery and/or recycling (3R) |
| | | C4 | Disposal |

(2) System Inventory. In this stage, the data required for the definition of the EI of the products and processes involved in the predefined scope are acquired either first hand from the site or through EI declarations or databases such as Ecoinvent and ELCD [96]. (3) The third stage it to multiply the inventory data by the functional unit specified and determine the impact using environmental indicators [1]. Mostly, the following eight midpoint environmental categories are considered: Abiotic Depletion Potential ADP, Acidification Potential AP, Eutrophication Potential EP, Global Warming Potential GWP, Photochemical Ozone CPOCP, and consumption of primary energy, renewable (PE-RE) and non-renewable (PE-NRe). The former six categories are quantified using the CML baseline method, while the Cumulative Energy Demand method for the last two [97]. (4) The final stage is to analyse the concluded data and communicate it with the stakeholders.

Evangelista and de Brito [98] found that the EI (ADP, GWP, ODP, AP, EP and POCP) decrease between 6% and 8% and between 19% and 23%, when 30% and 100% of fine NA are replaced with RFA, respectively. According to these studies [99,100], impacts from NAC and RAC can be similar when the additional cement content of RAC is below 10%. However, these may not be a sustainable strategy because the EI of cement is significantly higher than that of the aggregates [101]. Estanqueiro et al. [2] carried out a calculation of the EI of NA (Scenario i) and RA in the manufacture of concrete using, for the latter, a recycling fixed (Scenario ii) and mobile plant (Scenario iii) and concluded that the use of RA in the production of concrete is more favourable than the use of NA only in terms of land use and respiratory inorganic-impact categories, resulting mainly from the exploitation of the quarry. This study also concluded, however, that coarse RA can present a better EI than coarse NA if fine RFA were also used in concrete production instead of being sent to a landfill. Additionally, they also reported that advantage of RFA in terms of EI are mainly dependent on transportation distances. The summary of the literature findings of the EI of RFA concrete compared to NA concrete could be seen in Table 3.

**Table 3.** Impact assessment values for producing 1 kg of different aggregates types.

| Source | Country | ADP | GWP | ODP | POCP | AP | EP | Pe-NRe |
|---|---|---|---|---|---|---|---|---|
| | | kg Sb eq | kg CO$_2$ eq | kg CFC$^{-11}$ eq | kg C$_2$H$_4$ eq | kg SO$_2$ eq | kg PO$_4^{-3}$ eq | MJ |
| **Natural Fine Aggregates** | | | | | | | | |
| Braga [102] | Portugal | $3.37 \times 10^{-10}$ | $9.87 \times 10^{-3}$ | $1.71 \times 10^{-11}$ | $2.80 \times 10^{-6}$ | $4.58 \times 10^{-5}$ | $1.08 \times 10^{-5}$ | $1.35 \times 10^{-1}$ |
| | | $1.24 \times 10^{-9}$ | $2.79 \times 10^{-2}$ | $2.26 \times 10^{-10}$ | $9.06 \times 10^{-6}$ | $1.59 \times 10^{-4}$ | $3.54 \times 10^{-5}$ | $3.92 \times 10^{-1}$ |
| | | $1.09 \times 10^{-9}$ | $2.44 \times 10^{-2}$ | $2.43 \times 10^{-10}$ | $7.83 \times 10^{-6}$ | $1.44 \times 10^{-4}$ | $3.18 \times 10^{-5}$ | $3.44 \times 10^{-1}$ |
| | | $1.39 \times 10^{-9}$ | $3.14 \times 10^{-2}$ | $2.09 \times 10^{-10}$ | $1.03 \times 10^{-5}$ | $1.75 \times 10^{-4}$ | $3.90 \times 10^{-5}$ | $4.41 \times 10^{-1}$ |
| Tošić et al. [103] | Serbia | | $1.43 \times 10^{-3}$ | | $2.78 \times 10^{-7}$ | $1.64 \times 10^{-5}$ | $2.02 \times 10^{-6}$ | $1.48 \times 10^{-05}$ |
| | | | $2.12 \times 10^{-3}$ | | $4.15 \times 10^{-7}$ | $2.42 \times 10^{-5}$ | $3.01 \times 10^{-6}$ | $2.19 \times 10^{-5}$ |
| Korre and Durucan [16] | UK | | $9.30 \times 10^{-4}$ | $1.06 \times 10^{-10}$ | $4.58 \times 10^{-7}$ | $5.85 \times 10^{-6}$ | $4.35 \times 10^{-7}$ | |
| | | | $3.29 \times 10^{-3}$ | $4.50 \times 10^{-10}$ | $1.20 \times 10^{-6}$ | $1.89 \times 10^{-5}$ | $1.07 \times 10^{-6}$ | |
| | | | $2.16 \times 10^{-3}$ | $3.19 \times 10^{-10}$ | $7.35 \times 10^{-1}$ | $1.20 \times 10^{-5}$ | $6.87 \times 10^{-7}$ | |
| | | | $1.85 \times 10^{-3}$ | $2.14 \times 10^{-10}$ | $9.85 \times 10^{-7}$ | $1.03 \times 10^{-5}$ | $5.90 \times 10^{-7}$ | |
| | | | $3.79 \times 10^{-2}$ | $8.50 \times 10^{-6}$ | $5.40 \times 10^{-5}$ | $6.77 \times 10^{-4}$ | $1.04 \times 10^{-4}$ | |
| | | | $3.80 \times 10^{-2}$ | $1.78 \times 10^{-10}$ | $5.40 \times 10^{-5}$ | $6.77 \times 10^{-4}$ | $1.04 \times 10^{-4}$ | |
| Marinkovic' et al. [18] | Serbia | | $1.43 \times 10^{-3}$ | | $2.82 \times 10^{-7}$ | $1.64 \times 10^{-5}$ | $2.02 \times 10^{-6}$ | |
| Sjunnesson [104] | Sweden | | $1.60 \times 10^{-3}$ | | $1.70 \times 10^{-6}$ | $7.80 \times 10^{-7}$ | | $3.00 \times 10^{-2}$ |
| | | | $7.00 \times 10^{-4}$ | | $3.80 \times 10^{-10}$ | $5.00 \times 10^{-5}$ | | $1.24 \times 10^{-3}$ |
| Average | | $1.01 \times 10^{-9}$ | $1.23 \times 10^{-2}$ | $8.50 \times 10^{-7}$ | $4.90 \times 10^{-2}$ | $1.36 \times 10^{-4}$ | $2.58 \times 10^{-5}$ | $1.68 \times 10^{-1}$ |
| **Recycled Fine Aggregates** | | | | | | | | |
| Braga [102] | Portugal | $2.12 \times 10^{-10}$ | $7.44 \times 10^{-3}$ | $1.60 \times 10^{-10}$ | $2.14 \times 10^{-6}$ | $4.05 \times 10^{-5}$ | $9.28 \times 10^{-6}$ | $1.08 \times 10^{-1}$ |
| Tošić et al. [103] | Serbia | | $2.28 \times 10^{-3}$ | | $7.03 \times 10^{-7}$ | $2.49 \times 10^{-5}$ | $3.01 \times 10^{-6}$ | $2.59 \times 10^{-5}$ |
| | | | $3.38 \times 10^{-3}$ | | $1.18 \times 10^{-6}$ | $3.61 \times 10^{-5}$ | $4.34 \times 10^{-6}$ | $3.95 \times 10^{-5}$ |
| Korre and Durucan [16] | UK | | $2.42 \times 10^{-3}$ | $2.83 \times 10^{-10}$ | $8.00 \times 10^{-7}$ | $1.21 \times 10^{-5}$ | $7.06 \times 10^{-7}$ | |
| Marinkovic´ et al. [18] | Serbia | | $1.74 \times 10^{-3}$ | | $3.40 \times 10^{-7}$ | $2.00 \times 10^{-5}$ | $2.47 \times 10^{-6}$ | |
| Average | | $2.12 \times 10^{-10}$ | $3.45 \times 10^{-3}$ | $2.22 \times 10^{-10}$ | $1.03 \times 10^{-6}$ | $2.67 \times 10^{-5}$ | $3.96 \times 10^{-6}$ | $3.60 \times 10^{-2}$ |

The EI of producing aggregates could be attributed to two main processes; the energy of extracting natural aggregates versus recycling CDW in addition to the transportation to the concrete batch plant. Although the EI assessment data vary widely, there is a unanimous agreement in the references that the EI of recycled aggregates in concrete is proportional to the transportation distances whether a mobile or a normal stationary plant was used (since the distance of transporting CDW would still be the same). Marinkovic' et al. [18] compared the EI of two types of concrete mixes, one using NA and the other using RA, in the Serbian context. The study was based on two transportation scenarios: the first had the concrete batch plant only 15 km away from the recycling plant for RA, while the second considered it as 100 km. The study concluded that the EI of NA and RAC are mostly dependent on travel-distances and transport-type of aggregates between construction sites and recycling plants. Moreover, when transport distances of RA are smaller than that of NA, the EI of the resulting RAC is less than that of NAC even if the former had 3% extra cement in the mix [18].

It is known that the transportation distance changes between the supplier of raw material and the concrete plant based on the region considered. Therefore, a sensitive analysis is recommended before considering the transportation scenario. For example, the method developed by study of Göswein et al. [105] can be used to obtain a sensitive analysis. According to ELCD core database V3.0, two main types of lorries (medium sized lorry transport, low impact, maximum capacity of 17.3 tonnes; articulated lorry transport: high impact, maximum capacity of 27 tonnes) are used to transport raw materials to concrete plant (Table 4).

**Table 4.** Impact−assessment results to transport one kg·km (ELCD-core database V3.0).

| Lorry/Maximum Capacity (Tonnes) | Baseline CML Method | | | | | | Cumulative Energy Demand |
|---|---|---|---|---|---|---|---|
| | ADP | GWP | ODP | POCP | AP | EP | P X10-NRe |
| | kg Sb eq | kg CO$_2$ eq | kg CFC$^{-11}$ eq | kg C$_2$H$_4$ eq | kg SO$_2$ eq | kg PO$_4$$^{-3}$ eq | MJ |
| Articulated-lorry transport/27 t | $1.98 \times 10^{-12}$ | $4.98 \times 10^{-5}$ | $1.01 \times 10^{-13}$ | $1.59 \times 10^{-8}$ | $2.24 \times 10^{-7}$ | $5.14 \times 10^{-8}$ | $6.73 \times 10^{-4}$ |
| Lorry−transport/17.3 t | $2.62 \times 10^{-12}$ | $6.57 \times 10^{-5}$ | $1.33 \times 10^{-13}$ | $2.24 \times 10^{-8}$ | $3.11 \times 10^{-7}$ | $7.20 \times 10^{-8}$ | $9.27 \times 10^{-4}$ |

Accordingly, it could be concluded that, as shown in Table 5, the effect of the incorporation % of RFA on the resulting EI of concrete is linear to the difference in transportation distances between natural aggregates and RFA. The equations for the different indicators Y could now be calculated as the impact for 1 kg of fine aggregates concrete based on % incorporation (X) of RFA where D1 and D2 are the distance of transportation in km for natural aggregates and RFA, respectively.

**Table 5.** Predictive models of different mid-point environmental indicators for the % of incorporation X of RFA in concrete.

| Environmental Indicators | | NA (Average) | NA/km | RFA (Average) | RFA/km | Equation for Y (Impact for 1 kg of Fine Aggregates Concrete Based on % Incorporation (X) of RFA |
|---|---|---|---|---|---|---|
| ADP | kg Sb $_{eq}$ | $1.01 \times 10^{-9}$ | $9.47 \times 10^{-12}$ | $2.12 \times 10^{-10}$ | $1.70 \times 10^{-12}$ | Y = (D2 $\times$ 1.7 $\times 10^{-12}$ − D1 $\times$ 9.47 $\times 10^{-12}$) X + D1 $\times$ 9.47 $\times 10^{-12}$ |
| GWP | kg CO$_2$ $_{eq}$ | $1.23 \times 10^{-2}$ | $1.15 \times 10^{-4}$ | $3.45 \times 10^{-3}$ | $2.76 \times 10^{-5}$ | Y= (D2 $\times$ 2.76 $\times 10^{-5}$ − D1 $\times$ 1.15 $\times 10^{-4}$) X + D1 $\times$ 1.15 $\times 10^{-4}$ |
| ODP | kg CFC$^{-11}$ $_{eq}$ | $8.50 \times 10^{-7}$ | $7.97 \times 10^{-9}$ | $2.22 \times 10^{-10}$ | $1.78 \times 10^{-12}$ | Y = (D2 $\times$ 1.78 $\times 10^{-12}$ − D1 $\times$ 7.97 $\times 10^{-9}$) X + D1 $\times$ 7.97 $\times 10^{-9}$ |
| POCP | kg C$_2$H$_4$ $_{eq}$ | $4.90 \times 10^{-2}$ | $4.59 \times 10^{-4}$ | $1.03 \times 10^{-6}$ | $8.24 \times 10^{-9}$ | Y = (D2 $\times$ 8.24 $\times 10^{-9}$ − D1 $\times$ 4.59 $\times 10^{-4}$) X + D1 $\times$ 4.59 $\times 10^{-4}$ |
| AP | kg SO$_2$ $_{eq}$ | $1.36 \times 10^{-4}$ | $1.28 \times 10^{-6}$ | $2.67 \times 10^{-5}$ | $2.14 \times 10^{-7}$ | Y= (D2 $\times$ 2.14 $\times 10^{-7}$ − D1 $\times$ 1.28 $\times 10^{-6}$) X + D1 $\times$ 1.28 $\times 10^{-6}$ |
| EP | kg PO$_4$$^{-3}$ $_{eq}$ | $2.58 \times 10^{-5}$ | $2.42 \times 10^{-7}$ | $3.96 \times 10^{-6}$ | $3.17 \times 10^{-8}$ | Y= (D2 $\times$ 3.17 $\times 10^{-8}$ − D1 $\times$ 2.42 $\times 10^{-7}$) X + D1 $\times$ 2.47 $\times 10^{-7}$ |
| PE-NRe | MJ | $1.68 \times 10^{-1}$ | $1.58 \times 10^{-3}$ | $3.60 \times 10^{-2}$ | $2.88 \times 10^{-4}$ | Y = (D2 $\times$ 2.88 $\times 10^{-4}$ − D1 $\times$ 1.58 $\times 10^{-3}$) X + D1 $\times$ 2.88 $\times 10^{-3}$ |

*2.4. The Effect of Varying RFA Incorporation Ratio on Economic Impact of Concrete*

It is significant to consider the economic impact of a material if the objective is to assess its sustainability. The cost of construction materials is the second most influential factor (after functionality) in the decision-making process regarding selection of alternatives [25]. According to Hafez et al. [95], the eco-costs/value ratio of conventional concrete could be compared to that of other Green concrete mixes on the bases of the net present value considering that the study takes into consideration a replacement ration (N) that compares the service life of the alternatives being considered as well as the interest rate and the expected inflation rate during the time period of the forecasted cashflow [97]. A simpler way of quantifying the economic burden of concrete is to calculate its market price and compare it to the other alternatives using Equation (6) [106].

$$\text{Cost of concrete} \quad \text{alternative per unit volume (x)} = \sum_{Y=1}^{n} mass\ of\ constituent\ Y\ per\ unit\ volumen * market\ price\ of\ Y\ per\ unit\ mass \tag{6}$$

Braga [102] concluded that using limestone instead of granite aggregates results in 50% cost reduction while using recycled aggregates results in 80% savings. However, similar to the findings from 2.3, the economics of replacing sand with RFA was found to be linearly proportional with the

transportation distances. Since the scope of this paper is limited to fine aggregate replacement, the following equation could be used to estimate the cost instead:

$$\text{Cost of concrete} \quad \text{constituent } (Y) = \textit{Cost of raw materials production} + \textit{Cost of raw materials transportation} \tag{7}$$

According to Tošić et al. [103], the cost of RFA and sand excluding transportation is 11.5 and 3 €/tonne, respectively, while the cost of transportation is 0.02 €/tonne/km. Yang et al. [107] cited the transportation cost of aggregates as 0.08 €/tonne/km so the average taken will be 0.05 €/tonne/km. Moreover, Braga et al. [102] noted the cost of RFA and sand excluding transportation is 4.5 and 4 €/tonne, respectively, so the average for that is taken in this study as 8 €/tonne for RFA and 3.5 €/tonne for sand. It is important to state that in all the calculations of environmental and economic impact, the transportation means is assumed to be a large truck. For the environmental impact, the distances are assumed to be double the actual geographic distances to allow for an empty return trip but not for the economic impact calculations. Hence, the following equation could be descriptive of the cost (Y) of a tonne of the fine aggregate portion of the concrete alternative studied with the incorporation % X of RFA where D1 and D2 are the distance of transportation in km for natural aggregates and RFA, respectively, ignoring the economy of scale and the variability in market prices (Equation (8))

$$Y_{\text{economic}} = 3.5 \times D1 + X \times (0.045 \times D2 - 3.5 \times D1) \tag{8}$$

## 3. Sustainability Assessment of RFA Incorporation Ratio Based on Multi Criteria Decision Analysis

The objective of this section is to generate a framework for assessing the optimum replacement ratio of sand or natural fine aggregates with RFA from CDW. A multi criteria decision analysis framework was developed by Mateus et al. [108] as shown in Figure 13, which combines the functional, economic and environmental parameters of concrete to assess its sustainability. This is achieved in four main steps: First, the alternatives that are to be compared need to be defined. Second, based on this data, the different alternatives are evaluated for their functional, environmental and economic parameters. Third, these impact factors would be normalized according to the equation below for a factor between 0 and 1. Finally, based on user-defined weights, these three factors are to be aggregated together to form one sustainability index upon which the alternatives are to be ranked, and the optimal one is selected.

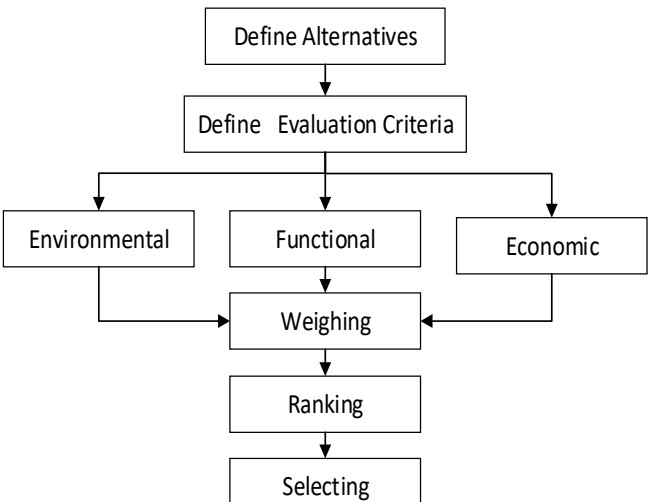

**Figure 13.** The support method for the assessment of the relative sustainability of building technologies proposed multi criteria decision analysis framework for sustainability assessment.

In light of the presented findings from the literature, the selected parameters and corresponding predictive models were summarized in Table 6. The authors of this paper chose to include only a selected few functional and environmental properties of concrete but rather followed the trend set by similar studies such as Silgado et al. [9] and Rahla et al. [109].

**Table 6.** A summary of the selected parameters, their weights and the regression models generated form the literature.

| Parameter | Empirical Function of Incorporation Ratio X (%) for the RFA in the Concrete Mix | Weights (%) | |
|---|---|---|---|
| Y (Functional) | $Y_{fc} = F_{ci} - 0.022X$ <br> $Y_E = 2.58 - 0.0159F_c^2 + 1.9107F_c - 13.276$ <br> $Y_{CD} = D_{90D} + 0.041X$ <br> $Y_{cl} (P_{Cl2}/P_{Cl1}) = 1 + 0.0033X$ | 25 <br> 25 <br> 25 <br> 25 | 33.3 |
| Y (Environmental) | $Y_{ADP} = (D2 \times 1.7 \times 10^{-12} - D1 \times 9.47 \times 10^{-12}) X + D1 \times 9.47 \times 10^{-12}$ | 14.3 | 33.3 |
| | $Y_{GWP} = (D2 \times 2.65 \times 10^{-5} - D1 \times 1.16 \times 10^{-4}) X + D1 \times 1.16 \times 10^{-4}$ | 14.3 | |
| | $Y_{ODP} = (D2 \times 1.77 \times 10^{-12} - D1 \times 2 \times 10^{-12}) X + D1 \times 2 \times 10^{-12}$ | 14.3 | |
| | $Y_{POCP} = D2 \times 8.26 \times 10^{-9} - D1 \times 9.42 \times 10^{-8}) X + D1 \times 9.42 \times 10^{-8}$ | 14.3 | |
| | $Y_{AP} = (D2 \times 2.14 \times 10^{-7} - D1 \times 1.27 \times 10^{-6}) X + D1 \times 1.27 \times 10^{-6}$ | 14.3 | |
| | $Y_{EP} = (D2 \times 3.17 \times 10^{-8} - D1 \times 2.41 \times 10^{-7}) X + D1 \times 2.41 \times 10^{-7}$ | 14.3 | |
| | $Y_{Pe-NRe} = (D2 \times 2.88 \times 10^{-4} - D1 \times 1.57 \times 10^{-3}) X + D1 \times 1.57 \times 10^{-3}$ | 14.3 | |
| Y (Economic) | Yeconomic $= 3.5 \times D1 + X \times (0.045 \times D2 - 3.5 \times D1)$ | 100 | 33.3 |

Three scenarios were assumed for the transportation distances since it proved as a vital variable in the comparison, especially in the environmental and economic impact. The assumed values for D1, the transportation distance between the natural aggregates source and the concrete batch plan, were 150, 100 and 50 km. Similarly, the values for D2, the distance between RFA and the concrete batch plan, were assumed as 50, 100 and 150 km, in that order.

The conventional concrete mix was assumed to have the following functional characteristics with zero RFA added:

- Compressive strength = 30 MPa
- Expected maximum carbonation Depth = 20 mm
- Permeability to chlorides = $10 \times 10^{-12}$ m$^2$/s

The 11 alternatives representing possible values for X were modelled using the framework described and the results were summarized in Table 7 as follows.

**Table 7.** A summary of the sustainability assessment using the multicriteria decision analysis framework.

| Alternatives | X (%) | Functional | Scenario 1 | | | Scenario 2 | | | Scenario 3 | | |
|---|---|---|---|---|---|---|---|---|---|---|---|
| | | | Environmental | Economic | Single Score | Environmental | Economic | Single Score | Environmental | Economic | Single Score |
| NA | 0% | 1.00 | 0.13 | 1.00 | 0.71 | 0.27 | 1.00 | 0.76 | 0.43 | 1.00 | 0.81 |
| 1 | 10 | 0.98 | 0.14 | 0.98 | 0.70 | 0.28 | 0.89 | 0.72 | 0.44 | 0.78 | 0.74 |
| 2 | 20 | 0.97 | 0.16 | 0.96 | 0.69 | 0.30 | 0.81 | 0.69 | 0.47 | 0.64 | 0.69 |
| 3 | 30 | 0.95 | 0.17 | 0.93 | 0.69 | 0.32 | 0.74 | 0.67 | 0.49 | 0.54 | 0.66 |
| 4 | 40 | 0.93 | 0.19 | 0.91 | 0.68 | 0.34 | 0.68 | 0.65 | 0.52 | 0.47 | 0.64 |
| 5 | 50 | 0.92 | 0.22 | 0.89 | 0.68 | 0.37 | 0.63 | 0.64 | 0.56 | 0.42 | 0.63 |
| 6 | 60 | 0.90 | 0.25 | 0.88 | 0.68 | 0.41 | 0.58 | 0.63 | 0.61 | 0.37 | 0.63 |
| 7 | 70 | 0.89 | 0.30 | 0.86 | 0.68 | 0.47 | 0.54 | 0.63 | 0.66 | 0.34 | 0.63 |
| 8 | 80 | 0.87 | 0.38 | 0.84 | 0.70 | 0.55 | 0.51 | 0.65 | 0.74 | 0.31 | 0.64 |
| 9 | 90 | 0.86 | 0.52 | 0.83 | 0.73 | 0.70 | 0.48 | 0.68 | 0.84 | 0.28 | 0.66 |
| 10 | 100 | 0.84 | 1.00 | 0.81 | 0.88 | 1.00 | 0.45 | 0.76 | 1.00 | 0.26 | 0.70 |

The results present enough evidence to support a claim that the incorporation of any increment of RFA to replace natural fine aggregates has an adverse effect on the overall functional performance of concrete. Nevertheless, it could be also concluded as per the comparison shown in Figure 14, regardless of the transportation distance, and due to the much higher purchase cost assumed for RFA, any incorporation % would yield a more expensive concrete alternative.

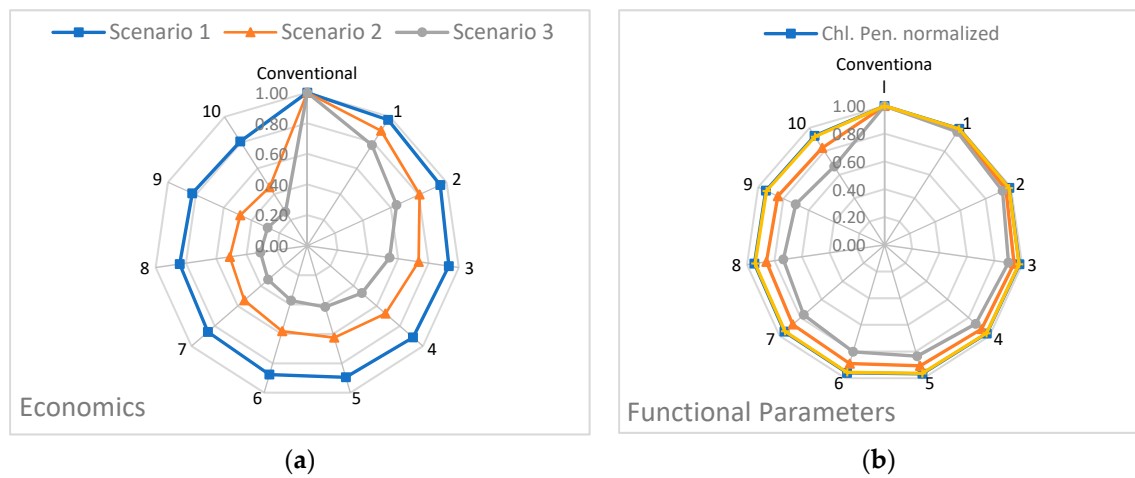

**Figure 14.** A comparison between the normalized values for the (**a**) economic and (**b**) functional parameters of the RFA based concrete alternatives.

On the other hand, the EI of the studied alternatives incorporating a range of RCA% (10% to 100%) showed sensitivity to the relative transportation distance of natural aggregates and RFA. Agreeing with the aforementioned data from the literature, when the distances travelled by NA double that assumed for RFA, which is described in this study as scenario 1, the concrete alternatives with RFA showed enhance environmental performance. As shown in Figure 15, mid-point indicators such as abiotic depletion potential (ADP) and the consumption of non-renewable primary energy (Pe-NRe) testify that alternative 10, which suggests the full incorporation of RFA instead of natural sand, is 80%, 60% and 40% better after normalization compared to the NA conventional concrete alternative considering scenarios 1, 2 and 3, respectively.

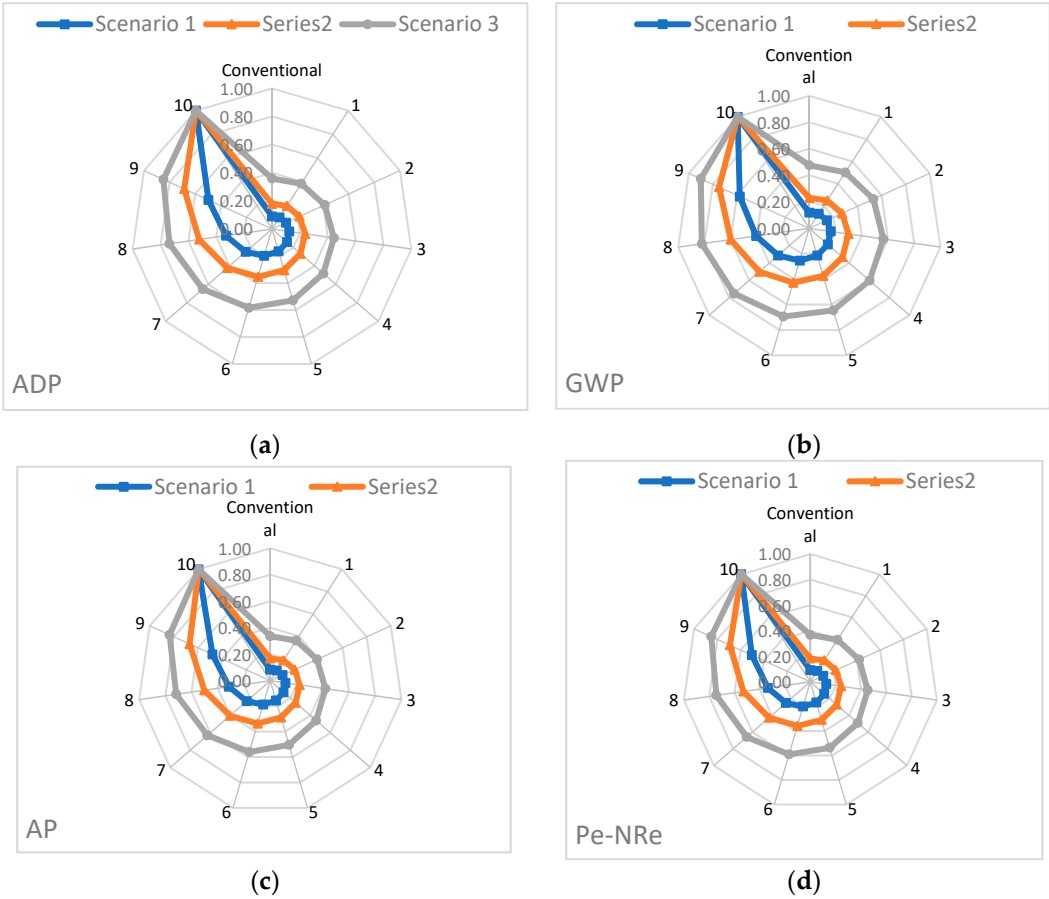

**Figure 15.** EI assessment of the proposed scenarios and RFA concrete alternatives using the following Mid-point indicators: (**a**) ADP, (**b**) GWP, (**c**) AP and (**d**) Pe-NRe.

## 4. Conclusions

This study aimed at investigating, through an extensive literature review, the performance of concrete in which RFA from CDW are used to replace, partially or totally, natural fine aggregates. The aim was to model the combined functional, environmental and economic performance parameters of different incorporation levels of RFA to judge the optimum value that achieves sustainability of concrete. A multi criteria decision analysis framework was applied, and several boundaries were assumed for the most significant variables: compressive strength, carbonation depth, chloride penetration and the transportation distances for the aggregates. Results showed that due to the significant decrease in the functional and economic parameters of concrete with any level of RFA incorporation, it is only possible to achieve a more overall sustainable concrete mix when the transportation distance of natural aggregates is at least double that of the RFA.

**Author Contributions:** Conceptualization, R.K. (Rawaz Kurda); methodology, R.K. (Rawaz Kurda); investigation, H.H. and R.K. (Rawaz Kurda); resources, R.K. (Rawaz Kurda); data curation, R.K. (Rawaz Kurda), H.H. and R.K. (Reben Kurda); writing—original draft preparation, R.K. (Rawaz Kurda) and H.H.; writing—review and editing, H.H., R.K. (Rawaz Kurda), R.K. (Reben Kurda) and B.A.-H.; visualization, H.H, R.K. (Reben Kurda), R.K. (Rawaz Kurda), B.A.-H., R.M. and B.A.; supervision, H.H., R.K. (Reben Kurda), R.K. (Rawaz Kurda), B.A.-H., R.M. and B.A.; project administration, H.H., R.K. (Reben Kurda), R.K. (Rawaz Kurda), B.A.-H., R.M. and B.A. In addition, data analysis, computing and optimization procedure were conducted by R.K. from the department of Information Systems Engineering, Erbil Technical Engineering College, Erbil Polytechnic University. All authors have read and agreed to the published version of the manuscript.

**Funding:** This work received no funding.

**Conflicts of Interest:** The authors declare no conflict of interests.

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
