# Peer review of "A Critical Review on the Influence of Fine Recycled Aggregates on Technical Performance, Environmental Impact and Cost of Concrete"

_applsci, doi:10.3390/app10031018_

Round 1

Reviewer 1 Report

The aim of this paper is to show the applicability of recycled fine aggregates (RFA) in concrete regarding the technical performance, environmental impact, energy consumption and cost. 

It is a revison paper about the state of the art of the use of RFA. No novelty is present in this work, however is a Good Review Report with valid conclusions. "Results showed that the due to the significant decrease 
 in the functional and economic parameters of concrete with any level of RFA incorporation, it is only possible to achieve a more overall sustainable concrete mix when the transportation distance of natural aggregates is at least double that of the RFA". I recommed introduce more information about CO2 pollution...in transport.

Quality of some figures (mainly,  figure 3) must be improved.

Quality of equation 3 must be improved.

Some references are old. Please, introduce new references.

Author Response

Enclosed please find the revised version of the manuscript and our feedback to the reviewer's comment.

Reviewer 2 Report

The review entitled “A critical review on the influence of fine recycled aggregates on technical performance, environmental impact and cost of concrete”.

The aim of the review was to analyze effects of incorporate recycled fine aggregates (RFA) in concrete production on the functional properties of concrete, the environment, and the cost-effectiveness of concrete production. This is a relevant topic for analysis.

However, this review is based on "old" literature.

The section State-of-art review contains two typical statistical errors: sacrificial pseudoreplication and desire to build a linear regression based on three experimental points.

The review also contains a huge number of structural, bibliographic and statistical errors, which are listed below.

The review contains a large number of abbreviations. This makes reading the article difficult. Abbreviations should be removed from the article as much as possible.

The review can be accepted after major revision.

Lines 118-124

The Geng and Sun [43] study does not contain the following sentence “…that larger RFA particles have a higher adhesion properties due to the presence of a higher % of old adhered binder”.

The study by Geng and Sun [43] does not contain data from Figure 2.

Lines 126-132

The study of Soliman [39] should be excluded from the review, since Soliman's Ph.D. thesis (2005) has not been published. Therefore, based on this fact, Figure 3 should be deleted.

The Soliman [39] data should be deleted from the Figure 7 on the line 226 and Figure 6 on the line 251, Figure 7 on the line 263.

This review contains two tables 6, two tables 7. The numbering order of the figures must be restored.

Lines 145-146.

As noted above, the Geng and Sun (2013) study was numbered 43, while Kim et al. was also numbered 43. The order between references must be restored.

Line 149. Figure 4.

Data from Yaprak et al. (2011) (Figure 1), Zega and Di Maio (2011) (Table 3) and Geng and Sun (2013) (Table 4) were mistakenly copied. A critical reanalysis must be performed.

Slump should be presented in mm not cm.

Linear equations cannot be built on the basis of only three points (see Zega and Di Maio (2011) and Geng and Sun (2013)).

Line 158, equation 1.

Only Yaprak et al. (2011) results can be used for the equation 1, if the four Gauss–Markov assumptions were met.

Lines 160-180. Figure 5 contains five data sets from five original studies. Four original studies contain high- and normal-strength concrete data sets. Combining data from different sets into one set is a statistical error. This error is called sacrificial pseudoreplication (Hurlbert 2009).

Hurlbert, S.H. The ancient black art and transdisciplinary extent of pseudoreplication. J. Comp. Psychol. 2009, 123, 434–443.

Line 198, Figure 6 and line 217, Equation 2. Figure 6 illustrates a classical sacrificial pseudoreplication (see explanation to figure 5). A linear equation cannot be built if there is only three points.

Line 226, Figure 7. Figure 7 contains the same statistical error that was illustrated in Figures 5 and 6.

Line 251, Figure 6. Figure 6 contains the same statistical error that was illustrated in Figures 5 -7.

Line 263, Figure 7. Figure 7 contains the same statistical error that was illustrated in Figures 5 -7.

Line 274, Figure 8. Linear equations based only on three points. A linear equation cannot be built if there is only three points.

Line 300, Figure 9. Figure 9 contains the same statistical error that was illustrated in Figures 5 -7.

Lines 341-343.

The following two references are Weil et al. [77] (2006) and Knoeri et al. [78] (2013) were cited formally.

Weil et al. [77] does not contain the following term “steel scrap”. Knoeri et al. [78] used the term “recovered steel scrap” but this procedure is no related to the "recycled aggregates concrete".

The Weil et al. [77] and Knoeri et al. [78] studies should be reanalyzed and critical analysis of these studies should be rewritten.

Line 344.

Estanqueiro et al. [1] analyzed the relationship between coarse recycled aggregates and fine recycled aggregates in concrete production. This is directly related to the current review. However, this is not discussed in this review.

Line 350, Table 3.

Table 3 is not mentioned in the text of the article.

Average and standard deviation should be deleted because this is not scientific information.

Line 373, Table 4. Table 4 is erroneously mentioned in the text of the article.

Line 382. This sentence “The results showed that the use of RA instead of NA is very beneficial” must be confirmed by reference.

Line 408. It was written “as shown in Figure 14”. However, the manuscript does not contain Figure 14.

Figure 10 was located on line 416 and was mentioned on line 249. Possible Figure 14 is Figure 10.

Line 421. Table 5 contains false functional equations because experimental units from different experimental sets were treated as statistically independent units.

Line 437. It was written “shown in Figure 15”. However, the manuscript does not contain Figure 15.

Figure 11 was located on line 441 and was mentioned on line 252.

Line 447. It was written “As shown in Figure 16”. However, the manuscript does not contain Figure 14.

Figure 12 was located on line 454 and was mentioned on line 270.

Author Response

(The authors gave the same response as above.)

Reviewer 3 Report

Thank you for the interesting manuscript with a title: "A critical review on the influence of fine recycled aggregates on technical performance, environmental impact and cost of concrete”.
The paper writes very well including:
the presented work is a significant contribution to the analysed topic; work is very well organized and comprehensively described;
the presented material is scientifically sound.

Several suggestions and comments regarding the manuscript please find bellow:

The MCDA parameter Y (Economic) can by appraisal not only one but in several Empirical functions of cost.

The newly literature source (2019-2020 year) regarding analysed problem can be presented in the review and the reference list.

In the paper similarity index 30%. Please find the results in the attachment. I recommended decreasing the duplication in the manuscript less 30%.

Reviewer

Author Response

(The authors gave the same response as above.)

Round 2

Reviewer 2 Report

The manuscript number: applsci-695569-peer-review-v2

A review of the manuscript titled “A critical review on the influence of fine recycled aggregates on technical performance, environmental impact and cost of concrete”.

Statistical analysis.

The authors used the results of regression equations from polling data to conduct a multi-criteria analysis of solutions. Unfortunately, today this is a common practice. However, using regression equations without checking statistical assumptions can lead to erroneous results.

It is well known that the model of the regression equation is based on the Ordinary Least Squares (OLS) method. To use the OLS method, the Gauss–Markov assumptions should be met earlier. In context of this manuscript (small sample sizes), the assumption of normality in residuals distribution is critical. To evaluate this assumption, the two-tailed Shapiro-Wilk test can be used.

This manuscript may be accepted after a minor change.

Figures and References.

There are errors in the numbering of the figures.

Line 183 Figure 5

Line 256 Figure 5

Line 214 Figure 6

Line 268 Figure 6

Line 243 Figure 7

Line 280 Figure 7

References 15 and 108 are one reference.

Relevance of the references should be checked.

Author Response

Enclosed please see the revised file and answers for the reviewer's comments.
